# Evaluation of the Physicochemical Properties of Chitosans in Inducing the Defense Response of *Coffea arabica* against the Fungus *Hemileia vastatrix*

**DOI:** 10.3390/polym13121940

**Published:** 2021-06-11

**Authors:** Julio César López-Velázquez, José Nabor Haro-González, Soledad García-Morales, Hugo Espinosa-Andrews, Diego Eloyr Navarro-López, Mayra Itzcalotzin Montero-Cortés, Joaquín Alejandro Qui-Zapata

**Affiliations:** 1Biotecnología Vegetal, Centro de Investigación y Asistencia en Tecnología y Diseño del Estado de Jalisco AC., Camino Arenero 1227, El Bajío, Zapopan 45019, Mexico; jucelopez_al@ciatej.edu.mx; 2Tecnología Alimentaria, Centro de Investigación y Asistencia en Tecnología y Diseño del Estado de Jalisco AC., Camino Arenero 1227, El Bajío, Zapopan 45019, Mexico; joharo_al@ciatej.edu.mx (J.N.H.-G.); hespinosa@ciatej.mx (H.E.-A.); 3Biotecnología Vegetal, CONACYT-Centro de Investigación y Asistencia en Tecnología y Diseño del Estado de Jalisco AC., Zapopan 45019, Mexico; smorales@ciatej.mx; 4Escuela de Ingeniería y Ciencias, Tecnológico de Monterrey, General Ramón Corona 2514, Nuevo México, Zapopan 45201, Mexico; diegonl@tec.mx; 5Instituto Tecnológico de Tlajomulco/TecNM, Km. 10 Carretera Tlajomulco-San Miguel Cuyutlán, Tlajomulco de Zúñiga 45640, Mexico; mayra.mc@tlajomulco.tecnm.mx

**Keywords:** chitosan coffee rust, biopolymer, phenolic compounds elicitor, protection

## Abstract

Chitosan is a natural polymer, and its biological properties depend on factors such as the degree of deacetylation and polymerization, viscosity, molecular mass, and dissociation constant. Chitosan has multiple advantages: it is biodegradable, biocompatible, safe, inexpensive, and non-toxic. Due to these characteristics, it has a wide range of applications. In agriculture, one of the most promising properties of chitosan is as an elicitor in plant defense against pathogenic microorganisms. In this work, four kinds of chitosan (practical grade, low molecular weight, medium molecular weight, and high-density commercial food grade) were used in concentrations of 0.01 and 0.05% to evaluate its protective effect against coffee rust. The best treatment was chosen to evaluate the defense response in coffee plants. The results showed a protective effect using practical-grade and commercial food-grade chitosan. In addition, the activity of enzymes with β-1,3 glucanase and peroxidase was induced, and an increase in the amount of phenolic compounds was observed in plants treated with high-molecular-weight chitosan at 0.05%; therefore, chitosan can be considered an effective molecule for controlling coffee rust.

## 1. Introduction

Chitosan is a polycationic biopolymer composed of glucosamine and N-acetyl-glucosamine units derived from chitin, which is the second-most abundant biopolymer in nature and a component of shells and the exoskeletons of crustaceans, insects, fungi, and yeasts [1]. There is abundant literature on its commercial development as a biomaterial for food and on its use in the pharmaceutical, medical, textile, agricultural, and environmental protection industries [2]. In agriculture, several studies reported that chitosan’s unique biological properties make it an essential component for a new generation of products that have low environmental impact. Chitosan inhibits the growth and development of different phytopathogens, including fungi, oomycetes, and bacteria [3]. In addition, it activates the immune system of plants, conferring protection against a broad spectrum of pathogens [4].

To decrease attacks by pathogens, plants possess chemical, physical, and enzymatic defensive responses. This immune system can be induced when a cell membrane’s pattern recognition receptors (PRRs) recognize pathogen-associated molecular patterns (PAMPs) during pathogenesis, resulting in the emergence of PAMP-activated immunity (PTI). However, pathogens have effectors that inhibit PTI, resulting in the emergence of disease. This process is termed effector-activated susceptibility (ETS). On the other hand, plants have evolved a multitude of strategies to produce resistance (R) proteins that recognize and inhibit pathogen effectors, thus restoring immunity. This process is referred to as effector-activated immunity (ETI) [5].

Inducing the defense response can be achieved by plant defense inducers or elicitors, which modulate some responses in the plants [6], and chitosan is in this group. This polymer is recognized as a microbe-associated molecular template (MAMP) because its structure is very similar to that of fungal cell wall fragments [7]. When recognized by the plant, it causes changes that include a massive complex of diverse biochemical reactions: oxidative burst reactions, occurrence of free radicals, fatty acid oxidation, membrane transmissibility changes, synthesis of proteins and other substances that strengthen the cell wall, phytoalexin formation, enzyme activation, especially oxidoreductases (peroxidases, lipoxygenases, polyphenol oxidases) and phenylalanine ammonium lyase, and the synthesis of pathogen-induced proteins and protease inhibitors [8]. Importantly, host-stimulating compounds are produced at concentrations that effectively inhibit the ability of pathogens to infect [9].

Chitosan has shown its activity as a defense inducer in crops such as rubber trees [5], ginger [9], eggplant [10], tea plants [11], grape vines [12], and *Arabidopsis thaliana* [13], activating plant defense mechanisms and protecting against pathogens.

The efficiency of chitosan on disease control depends on the dose and virulence of the pathogen [1]. However, the degree of polymerization, molecular weight, chemical composition, and environmental conditions are the critical factors [4,14].

Coffee is the second-most traded commodity in the world next to oil. It is estimated that 125 million people make their living from coffee cultivation, including 25 million small producers. Almost all of the world’s coffee production comes tropical and subtropical zones, mostly in developing or underdeveloped countries. In many cases, coffee exports constitute a vital part of that country’s total income, and its production is a significant employment generator [15]. The plant species *Coffea arabica* is susceptible to the *Hemileia vastatrix* fungus, which is the causal agent of the most devastating phytopathogen, coffee rust [16]. The disease can cause a loss in yield between 35 and 50% if control measures are not taken when posthole formation on the leaves appears. A reduced rate of photosynthesis develops, causing premature leaf drop and resulting in dry branches [17]. Research into new means of coffee rust control is important, and chitosan is a viable option.

The objective of the present work was to evaluate different physicochemical properties of chitosan along with high-density food-grade chitosan for controlling *H. vastatrix* in coffee plants. The effect of the best performing chitosan in inducing a defense response to the *H. vastatrix–Coffea arabica* interaction was also evaluated.

## 2. Materials and Method

### 2.1. General Methods

#### 2.1.1. Location of the Study Area

The experiments were conducted at Unidad de Biotecnología Vegetal and Laboratorio Nacional PlanTECC of Centro de Investigación y Asistencia en Tecnología y Diseño del Estado de Jalisco, AC., Zapopan, Mexico.

#### 2.1.2. Plant Material

Six-month-old in vitro cultured and acclimatized *Coffea arabica* plants having six true leaves were used. Plant material was obtained from coffee seeds collected in the state of Chiapas, Mexico. For plant propagation, MS culture medium was used, which was supplemented with myo-inositol (Sigma-Aldrich, St. Louis, MO, USA)(100 mg/L), Morell’s vitamins (4 mg/L), cysteine HCl (Sigma-Aldrich, St. Louis, MO, USA) (25 mg/L), and sucrose (Sigma-Aldrich, St. Louis, MO, USA) (30 g/L) [18].

#### 2.1.3. Biological Material

The phytopathogenic fungus *Hemileia vastatrix,* isolated from infected coffee plants from the state of Colima, México was used. Fungal spores were taken from the infected leaves and resuspended in 0.05% (*v*/*v*) Tween 20, which was subsequently adjusted to a concentration of 1 × 10^5^ spores/mL [19]. The leaves were inoculated with 100 µL of the spore suspension. Finally, each plant was covered with a dark plastic bag to favor the germination of urediniospores. Plants were kept in a culture room at 26 ± 2 °C, with a light/dark photoperiod of 16/8 h, respectively.

#### 2.1.4. Chitosan Solutions

Practical-grade (PC), medium-molecular-weight (MC), and low-molecular-weight (LC) chitosan (Sigma-Aldrich, Darmstadt, Germany) and commercial high-density, food-grade chitosan (CQ, America Alimentos, Jalisco, México) were used.

### 2.2. Chitosan Characterization

#### 2.2.1. Degree of Deacetylation by Fourier Transform Infrared Spectroscopy (FTIR)

The FTIR spectra of solid samples of all four kinds of chitosan were obtained by using a FTIR GX System spectrophotometer (PerkinElmer, Shelton, CT, USA) coupled to an ATR DuraSample II accessory. All the spectra were from 4000 to 400 cm^−1^ at a resolution of 0.5 cm^−1^ [20,21]. The deacetylation degree was calculated using the following equations:DA (%) = (0.3822 − A_1320_/A_1420_)/0.03133(1)
DD (%) =100% − DA (%)(2)
where A_1320_ y A_1420_ represent the absorbance of the chitosan at 1320 and 1420 cm^−1^, respectively.

#### 2.2.2. Viscosity Average Molecular Weight Determination by Chitosan Intrinsic Viscosity

The average molecular weight (Mv) was obtained by determining the intrinsic viscosity (η). Viscosity measurements were carried out using the Ubbelhode 1C (CANNON Instrument Company, PA, USA). Chitosan samples were prepared using the 2% acetic acid and 0.2 M sodium acetate solvent system at 20 °C. Subsequently, different concentrations (1–10 mg/mL) were prepared. Solvent flowtime measurements were carried out 3 times. Average molecular weight viscosity was calculated using the Mark–Houwink–Sakurada equation, in which *k* and α are constants equal to 1.38 × 10^−2^ dm^3^/g and 0.85 respectively for the solvent system [20,22,23].
M*v* = ([η]/*k*)^(1/α)(3)

### 2.3. Evaluation Chitosan Protection in Coffee Rust Control

#### 2.3.1. Chitosan Solutions Preparations

Chitosan sources were initially diluted to 1% (*w*/*v*) in 0.4 M acetic acid. Subsequently, all chitosans were brought to a final concentration of 0.01 and 0.05% (*v*/*v*).

#### 2.3.2. Biological Effectiveness Test

An experiment was established with six-month-old coffee plants. These were treated with 3 mL solutions of practical-grade, medium-molecular-weight, low-molecular-weight, and high-density chitosan at 0.01 and 0.05% (*v*/*v*) [11,24] seven days prior to pathogen infection. Randomized blocks were performed with 10 replicates each including positive and negative controls (plants inoculated with only the pathogen and plants without any treatment, respectively). The evaluated parameters were disease incidence, severity, percentage of diseased leaves, and defoliation [17,25].

#### 2.3.3. Disease Incidence

Plants were visually evaluated, and disease incidence was calculated based on the plant tissue’s showing symptoms [26].
Disease incidence (%) = (Diseased plants)/(Total plants) × 100(4)

#### 2.3.4. Disease Severity

The 7-value scale proposed in the DGSV Technical Sheet No. 40 was used (Table 1) [27].

#### 2.3.5. Defoliation

Defoliation was determined using the following equation:Defoliation (%) = (Fallen leaves)/(Total leaves) × 100(5)

#### 2.3.6. Disease Leaves

Diseased leaves were determined using the following equation:Disease leaves (%) = (Diseased leaves)/(Total leaves) × 100(6)

#### 2.3.7. Area under the Curve Calculation for Disease Progression (AUDPC)

The area under the curve for disease progression was calculated using the following equation:AUDPC = ∑(i = 1)^n[(Y_i+1_) + Y_i_)/2] [X_i+1_ − X_i_](7)
where n is the number of disease measurements over time; (Y_i_ and Y_i+1_)/2 is the midpoint between (Y_i_ and Y_i+1_), representing the amount of disease in a time interval corresponding to the height of each rectangle; and (X_i+_ − X_i_) represents the time (d) between two disease assessments and the width of each rectangle [28].

### 2.4. Evaluation of Chitosan in the Defense Response in the C. arabica–H. vastatrix Interaction

An experiment was established with six-month-old coffee plants treated with 3 mL of practical-grade chitosan at 0.05% (*v/v*). The experiment was carried out under the same conditions described in the above experiment. The treatments considered were C (untreated plants), H (plants inoculated with the pathogen), and QH (practical-grade chitosan at 0.05% and pathogen inoculated).

#### 2.4.1. Enzymatic Extraction

Leaf samples were taken from plants at 0, 12, 24, 48, and 72 h after pathogen inoculation. Enzymatic activity determination from β-1,3-glucanase, peroxidase, and quantification of total phenolic compounds absorbed at 650 nm were evaluated. Samples were macerated with liquid nitrogen and resuspended in 0.1 M sodium phosphate buffer (pH 7.0) to determine β-1,3-glucanase and peroxidase activity and in 80% (*v/v*) methanol (Sigma-Aldrich, St. Louis, MO, USA) for the quantification of total phenolic compounds.

#### 2.4.2. β-1,3 Glucanase Activity Determination

The determination of pathogenesis-related (PR) proteins with β-1,3 glucanase (β1,3-G) activity was performed with the reducing sugar colorimetric detection method at 515 nm as an enzymatic hydrolysis product and DNS (3,5-dinitrosalicylic acid, Sigma-Aldrich, St. Louis, MO, USA) reduction. Quantification was determined using a glucose (Sigma-Aldrich, St. Louis, MO, USA) calibration curve, and activity was reported as nkat per gram of total protein (nkat/g TP). For this study, 1 nkat (nanokatal) was defined as 1 nmol of D-glucose released from laminarin per second under assay conditions [29].

#### 2.4.3. Peroxidase Activity Determination

Peroxidase activity (POX) quantification was performed in accordance with the method proposed by [29]. For the reactions, 10 µL of leaf and root extract were used. The variation of one absorbance unit per minute is defined as one unit of peroxidase activity (1 UA) and is expressed per gram of total protein (UA/g TP) [29].

#### 2.4.4. Total Phenolic Compounds Quantification

Total phenolic compound quantification was performed following the proposed methodology of [30] with some modifications. Fresh tissue (1 g) was macerated and resuspended in 5 mL of 80 % (*v/v*) methanol. The methanolic extract was used for quantification with Folin Ciocalteu’s reagent (MilliporeSigma, Darmstadt, Germany) for 5 min. Subsequently, the reactions were carried out in the presence of sodium carbonate 20% *v/v* (MilliporeSigma, St. Louis, MO, USA) and quantified at 650 nm. Quantification was performed using a standard curve of known catechol (Sigma-Aldrich, St. Louis, MO, USA)(0.2–1 mM) concentration, and the results were expressed as nmol catechol/mg FW [30].

### 2.5. Statistical Analysis

A single-factor analysis of variance was performed using Statgraphics Centurion XVI V16.1.11 (StatPoint Technologies, Inc., Warrenton, VA, USA). For a mean separation test, an analysis was performed using the LSD test with a 95% significance level.

## 3. Results

### 3.1. Chitosan Characterization

#### 3.1.1. Degree of Deacetylation

The chitosan sample FTIR spectra (Figure 1) exhibited characteristic absorption bands at 3360 cm^−1^ (O–H stretching), 1870–2880 cm^−1^ (CH_2_ stretching), 1645 cm^−1^ (C–O stretching), 1590 cm^−1^ (NH_2_ bending), 1420 cm^−1^ (CH_2_ and O-H bending), and 1320 cm^−1^ (C–N stretching and CH bending). The absorption bands at 1155 cm^−1^ (anti-symmetric stretching of the C–O–C bridge), 1060 cm^−1^, and 1030 cm^−1^ (C–OH stretching and CH bending, respectively) are characteristic of its saccharide structure [20,21].

Equations (1) and (2) were used to determine the degree of deacetylation of the chitosan samples (all over 75%) from the infrared spectra. The results are listed in Table 2.

#### 3.1.2. Molecular Weight Viscosity Average

The intrinsic viscosity (η) of the food-grade chitosan solution in 2% acetic acid and 0.2 M sodium acetate at 20 °C and the average molecular weight of viscosity (M*v*) are listed in Table 2. The food-grade chitosan has a low molecular weight because of the intrinsic viscosity of chitosan.

### 3.2. Chitosan Protection Evaluation in Coffee Rust Control

#### 3.2.1. Biological Effectiveness Test

During the first 10 days after inoculation with *H. vastatrix*, no changes were observed that would indicate whether the disease had become established, but at 40 days, symptoms appeared. Treatment H plants (inoculated with the pathogen and with a medium molecular weight chitosan treatment) showed a higher number of infected plants. The symptoms, according to the SENASICA 2019 scale, were at level 3, which indicated 7% leaf damage. The main symptoms were the appearance of small yellow spots on the underside of the leaves where the inoculation was carried out. In the case of treatment H, leaves showed chlorosis at the tip. The 0.05% low molecular weight chitosan treatment (LH5) showed 80% diseased plants, some of them with a severity degree of 4. The other treatments showed a percentage of disease incidence between 30 and 50% with symptoms between level 1 and 2, mainly isolated chlorotic spots and damage to 2% of the leaf. It is important to emphasize that the treatments of low-molecular-weight chitosan 0.01% (LH1) and food-grade chitosan 0.05% (CH5) showed a low percentages of incidence, but the plants that were evaluated up to day 40 showed grade 3 symptoms.

After 60 days, disease incidence was 100% for most treatments, except for the treatments of practical-grade chitosan 0.05% (PC5) and food-grade chitosan 0.01% (CH1), which had an incidence of 60%, indicating a delay in infection. Symptoms for most cases were located between infection grade 4 and 5, which were the H treatments. Plants treated with medium- and low-molecular-weight chitosan were the most affected: infected leaves exceeded 50%, and necrosis was observed at the leaf ends and toward the leaf center. Yellow spots, characteristic of the disease, increased in number, and the leaves began to lose firmness and turgor. In some cases where the plants showed more severe symptoms, the leaves detached from the stem at the slightest movement. The treatments that showed a low infection rate had tiny, scattered yellow spots, which were associated with infection grade 3.

After 80 days of inoculation with *H. vastatrix*, the disease incidence increased to 80% for plants treated with practical-grade chitosan 0.05% (PC5) and to 90% for food-grade chitosan. Symptoms for both were associated with grade 3 infection, but compared to the other treatments, infection was delayed. Therefore, they were less intense, and damage to the plant was 7%. The control plants did not show any symptoms of infection even when they were under the same conditions and beside the plants under evaluation. The symptoms in the treatments where infection had been present for days worsened. There was turgor loss in all treated plants and severe chlorosis in the diseased leaves. Treatments H, MH1, and MH5 were the most affected by this symptom, which appeared in all parts of the leaf. It should be noted that when the chlorosis appeared, it started at the ends of the leaves around where the yellow spots caused by the pathogen appeared. Furthermore, as the spots spread, they began to necrotize the leaves. These disease symptoms were associated with a severity grade 6, which is the maximum grade on the scale (Figure 2).

#### 3.2.2. Defoliation

One of the main symptoms of infection is defoliation, which was evident after 40 days. The most affected plants were those where the disease was established on the first day. No relationship was established between the amount of defoliation and treatment. Defoliation is a function of infected plants, and leaves fall by mechanical action when they turn necrotic; therefore, the amount of defoliation was not considered a relevant factor. In control plants, there was no defoliation due to infection, while the plants inoculated with the pathogen showed defoliation of 18.83%. In the treatments where disease incidence was lower or delayed, defoliation was very similar to the treatments that showed greater damage; however, only diseased leaves fell (Table 3).

#### 3.2.3. Disease Leaves

Although all leaves were inoculated, the disease was not present in the whole plant. This condition could have been due to the inoculated spore number per leaf or to the conditions in which they were kept. On the one hand, spores precipitate easily, and since the site of inoculation was the leaves’ underside, some drops with spores fell to other parts of the plant or went to the stem. The control treatment (C) showed no diseased leaves. The treatments that showed a lower percentage of diseased leaves and were statistically significant were PH5 (≈22%) followed by CH5 (≈24%). The most susceptible treatments were those of low- and medium-molecular-weight chitosan as well as CH1. According to disease incidence, these were the first treatments infected by *H. vastatrix*, and it can be inferred that the chitosan that was applied foliar before infection did not play an antimicrobial role. Although low-molecular-weight chitosan is known to be active against microorganisms, this effect was not observed in the present work (Table 3).

#### 3.2.4. Area under the Disease Progress Curve (AUDPC)

The AUDPC was determined based on the severity of the disease in the days evaluated, taking as a reference the scale proposed by SENASICA. The value was correlated with the treatments, and the result indicated that the higher the AUDPC value, the more susceptible the plant was to infection. Control plants remained at 0, and the highest value was for the H treatment, since it did not receive any preventive treatment to stop the infection; in turn, LH1, LH5, and MH1 showed the same statistical behavior, so they were susceptible to infection by *H. vastatrix;* in other words, low-molecular-weight and medium-molecular-weight chitosan did not have an effect on rust control. According to the AUDPC value, the treatments that worked to delay or prevent infection were PC5 and CH1, the values of which (≈75%) were lower, as shown in Table 3. This also agrees with the appearance of the disease, since these were the treatments in which the disease did not appear in its entirety. According to the infection severity scale, they were the treatments in which a lower degree of damage was observed (Table 3). With these observations, it could be inferred that practical-grade chitosan at a concentration of 0.05% is a promising molecule for use against coffee rust. Likewise, high-density food-grade chitosan can be considered; however, the concentration to be used in the latter case, and the applications to be made to the crop should be analyzed.

### 3.3. Chitosan Evaluation in the Defense Response in the C. arabica–H. vastatrix Interaction

#### 3.3.1. Enzyme Evaluation with β-1,3 Glucanase Activity

The activity of β-1,3 glucanase, presented in Figure 3a, was favored in the QH treatment; the priming effect was effective, since it showed a statistically significant difference, mainly during the 48 h after infection. The activity of these enzymes increased as the days passed and was maintained until 72 h with a value of 0.636 nkat/mg TP. In this case, the chitosan acted as a signaling agent of the infection, and when the pathogen was introduced, the plant response was enhanced. Concerning the control plants, the value increased 45.48, 56.88, and 29.99 times at 24, 48, and 72 h, respectively, suggesting that there had been an important and significant metabolic change in the plants that were treated and inoculated. As for treatment H, which did not receive preventive treatment, an increase in enzyme activity was observed at 12 h after inoculation with the pathogen, which even exceeded the priming treatment; however, this effect was inhibited at 24 h, when it behaved similarly to the control group. This effect decreased four times at 24 and 48 h after inoculation compared to the control group, so it is believed that the fungus managed to suppress the enzyme activity of, and was a factor in, the occurrence of infection.

#### 3.3.2. Peroxidase Activity Enzymes Evaluation

Peroxidase activity was observed to decrease at 12 h in all treatments. This may be because peroxidases played an important role during the establishment of the disease, and the fungus was able to suppress the response or because enzymes with peroxidase activity fulfilled their role during the first hours. In the evaluation, treatments C and H remained below the activities of the group of plants inoculated with the pathogen and treated with chitosan QH, which showed statistically significant activities. In contrast, at 24, 48, and 72 h after inoculation, the levels of enzyme activity showed statistically significant increases. In these cases, the level of activity varied between both treatments, having a higher activity in the treatments in which the induction with chitosan was performed. They showed values of 0.214, 0.209, and 0.467 UA/mg TP, at 24, 48, and 72 h, respectively (Figure 3b).

#### 3.3.3. Total Phenolic Compounds Quantification

Total phenolic compounds absorbing at 650 nm were quantified. A yield between 450 and 525 mg/FW was observed. Phenols at 650 nm were produced mainly in the first 24 h after infection; a greater amount was consistently produced in the treatment induced with chitosan and inoculated with the pathogen compared to the treatment inoculated only with the pathogen. However, the differences were not statistically significant until after 48 h, when the production peak of these compounds was the highest compared to all the other treatments and increased up to 24 times more than the control treatment. After 72 h, the content of phenolic compounds decreased in the treatments with the pathogen and became statistically equal (Figure 3c).

## 4. Discussion

The effect of chitosan having different physicochemical characteristics on the control of coffee yellow rust was evaluated, and it was observed that molecular weight and viscosity influenced the control of the disease. The results were favorable with practical-grade chitosan (PC) and high-density chitosan (CQ). Chitosan and oligomers are known to have different uses in various disciplines, including agriculture, to control pathogenic plant microorganisms, induce plant immune responses, or serve as a chelating agent to promote the uptake of nutrients and minerals [11]. The biological properties of chitosan largely depend on its polymerization degree (PD), deacetylation degree (DA), and viscosity [31,32]. In this case, chitosan application decreased the incidence of rust disease in infected plants and significantly reduced disease severity. The treatments that showed a reduction in the disease (the area under the disease progress curve, AUDPC) were the high concentration (PH5) of PC chitosan and the low concentration (CH1) of CQ chitosan. Although the other concentrations also showed a significant reduction, they were not higher than those of CH1 and PH5.

The other chitosans did not show a significant reduction in disease symptoms on coffee plants. According to their physicochemical properties, high-density chitosan (CQ) presented a high deacetylation degree (DD, 90.35%), an intermediate molecular weight (Mv, 314 kD), as well as a low intrinsic viscosity (η, 631). The PC chitosan presented a low degree of deacetylation (DD, 77.22%), a low molecular weight (Mυ, 260 kD), and a low intrinsic viscosity (η, 593). Of these physicochemical properties, the DD and molecular weight of the PC and CQ chitosans were not equivalent to each other, but their intrinsic viscosity was. In this work, viscosity made a significant difference in inducing protection against coffee rust. Of the three physicochemical properties considered important for the biological activity of chitosan as an elicitor, viscosity is the characteristic that has been least addressed in agricultural research [33,34]. Intrinsic viscosity (η) is a measure of the hydrodynamic volume occupied by macromolecules in solution and is therefore a reflection of their size. It is highly dependent on the concentration of the polymer in solution. This property is important for understanding the behavior of chitosan in solution and its rheology because films, membranes, fibers, and hydrogels are prepared from solutions of this polysaccharide [35]. The relationship between viscosity and antimicrobial activity of chitosan has been described in the literature. Khalil and Badawy worked with chitosan of different molecular weights (ranging from 2.27 to 9.47 × 10^5^ g/mol) and different viscosity ranges from 4.12 and 12.19), and they found that the nematicidal activity of chitosan increased as the molecular weight and viscosity decreased; furthermore, they indicated that as the molecular weight and viscosity increased, the nematicidal effect decreased considerably [36]. However, in the present study, it cannot be directly associated to an antimicrobial effect because the plants were pre-treated with chitosan prior to infection with coffee rust. This implies a defense induction effect mediated by defense inducers, where chitosan is considered to be the elicitor. A relevant point of viscosity is highlighted in the work of No et al., who evaluated the stability (via viscosity) of chitosan solutions under different storage times and conditions and their effect on the antimicrobial activity against Gram-positive (*Listeria monocytogenes* and *Staphylococcus aureus*) and Gram-negative (*Salmonella enteritidis* and *E. coli*) bacteria [37]. Their results showed that the viscosity of chitosan solutions decreases with increased storage time and temperature. Furthermore, they documented the instability of chitosan solutions under a temperature of 25 °C, changing their viscosity; however, they observed a decrease in antimicrobial activity in lower molecular weight chitosan (1110 kD) compared to higher molecular weight chitosan (2025 kD) [37]. In this study, the chitosan solutions that showed the greatest effect on disease control had intermediate molecular weights (260 kD for PC and 314 kD for CQ), which could be susceptible to changes in viscosity. However, there are a few reports associated with chitosan viscosity and its relationship with enhanced disease protection [12] as a plant defense inducer. Lucini et al. reported the use of low-viscosity chitosan and observed an increase in peroxidase activity and in triterpenoid production. This suggests that this physicochemical property of chitosan and its role in inducing plant defense and protection deserves further exploration, considering that it is a highly relevant property when developing a chitosan solution on a commercial level.

For plant defense induction, plants recognized chitosan as a microbe-associated molecular template (MAMP) because its structure is very similar to fungal cell wall fragments [7]. Therefore, biochemical and molecular changes were triggered, such as oxidative burst, cell wall lignification, callose deposition, increased PR proteins, hypersensitive response (HR), and phytoalexin accumulation [5,12]. In the *Coffea arabica–Hemileia vastatrix* interaction, the activity of β-1,3-glucanase decreased in those plants that showed symptoms of the disease compared to those that do not [38]. During the first days of infection, the activity of β-1,3-glucanase decreased but increased 10 to 40 days after fungal inoculation [39]. In the present work, practical-grade chitosan (PC) had an apparent resistance-inducing effect. During the first days after inoculation with coffee rust, an induction response was observed in the enzyme with β-1,3-glucanase activity, the function of which is to directly hydrolyze the cell wall of phytopathogenic fungi to release β-1,3-glucans and chitin oligosaccharides that stimulate host defense responses [9]. The effect of enzyme activity increased in coffee plants until the third day after inoculation. These results are similar to those obtained with other rust-control compounds, such as manganese phosphite, which is a compound reported to be an elicitor that causes the induction of genes related to β-1,3-glucanase (*GLU*) in coffee plants inoculated with the pathogen and previously treated with the compound [17]. A similar response was observed with epoxiconazole and pyraclostrobin, which are triazole fungicides attributed with antioxidant and photosynthetic protection that induce *GLU* gene expression during the first days of infection (39]. Additionally, the mycoparasite *Calonectria* of *Hemileia vastratix* triggers an elicitor response similar to those previously described in relation to *GLU* gene induction [40]. In coffee plants, the effect of chitosan on the activity of β-1,3-glucanase has not been described, but a similar response has been described in other pathosystems, as in the case of rubber trees and *Phytophthora palmivora*, where there was greater expression of the *GLU* gene and was maintained for up to 48 h [5]. In *Camellia sinesis*, an increase in enzymes with β-1,3-glucanase activity was observed during the first hours of exposure of plants treated with chitosan [11].

For the activity of peroxidases, in the *Coffea arabica–Hemileia vastatrix* interaction, they decreased in those plants that presented disease symptoms compared with those that did not [38]. During infection, peroxidase activity did not change during the first days, presenting peaks from 5 days after infection, decreasing after 10 days, and increasing again 26 days after inoculation (39). In this work, an increase in peroxidase activity was found from 24 h after pathogen inoculation and increased until 72 h, whereas untreated plants did not show an increase in peroxidase activity. The increase in peroxidase family enzymes may be associated with their involvement in the antioxidant system plant defense processes against pathogens, such as lignin formation, and the cross-linking of cell wall components to strengthen walls and limit pathogen invasion [41]. A similar response was reported for manganese phosphite, which caused the induction of peroxidase (POX)-related genes as soon as the compound was applied [17]. In addition, the induction response of antioxidant enzyme activity (APX, SOC, PPO) was similar, showing an increase 24 h after pathogen inoculation [17]. In contrast, for the fungicides epoxiconazole and pyraclostrobin, peroxidase activity was not induced and was always lower in infected plants [39]. In other pathosystems, the response induced by chitosan was similar to the results found in this work. For example, in *Vitis vinifera*, foliar application of 0.03% low-viscosity chitosan promoted the activity of copper and zinc superoxide dismutase (SOD) and glyoxal oxidase (GLOX) enzymes, thus activating defense and lignification processes in the hypersensitive response [12]. Enzymes with catalase (CAT), peroxidase (PO), and polyphenol oxidase (PPO) activity were increased during the first hours of infection in plants that had been treated with chitosan [12]. According to the results observed in this investigation and in agreement with those reported in other studies, peroxidase plays a vital role in the establishment of infection, i.e., during the first hours of exposure to the pathogen.

One of the main mechanisms of plant resistance to pathogen attack is through the production of secondary metabolites. Polyphenols are aromatic chemical compounds containing abundant phenolic groups that are divided into simple phenols, menolic acids, phenylpropenoids, coumarins, flavonoids, and tannins [42]. It has been reported that a higher proportion of polyphenols in coffee varieties may be associated with increased resistance to coffee leaf rust [42,43], although in susceptible varieties, there may be an increase in the content of total phenolic compounds during infection [38]. In this work, the application of PC chitosan increased the production of phenolic compounds after 48 h of pathogen inoculation, but this increase was not maintained after 72 h. Treatment with other elicitors, such as jasmonic acid, on coffee plants did not cause an increase in total phenolic compounds [43]. In other pathosystems, the response has been different. With the application of 1% low-molecular-weight chitosan to avocado plants, an induction of genes involved in phenylpropanoid biosynthesis was observed [7]. These results showed an increase in the concentration of these compounds, with the highest levels occurring during the first 48 h. Some authors reported that the duration of signal transduction and protective response induction mechanisms occurs between 24 and 96 h after the perception of biotic or abiotic stress. As phytoalexins can be phytotoxic, they must be metabolized rapidly in order not to remain in plant cells for a prolonged time [12]. Therefore, the fact that an increasing level of phenolic compounds was not maintained may not be related to an effective resistance response against the pathogen for a plant variety susceptible to coffee rust.

Due to the increase in phytosanitary problems to which plants are exposed and the need to minimize the use of chemical pesticides that cause damage to humans and the environment, ecological and economically attractive alternatives for the control of diseases caused by pathogens must be implemented. As observed in this work, elicitors or defense inducers, such as chitosan, are exogenous biological compounds capable of activating plant defense mechanisms against biotic or abiotic stress [12,32]. Although chitosan has characteristics that make it a strategic molecule with high potential for agricultural use, other aspects such as its low solubility and high viscosity need to be considered before obtaining chitosans or oligochitosans with characteristics that make them profitable and manageable for farmers.

## 5. Conclusions

Practical-grade (PC) and high-density chitosan (CQ) chitosan at concentrations of 0.05 and 0.01% decreased the incidence of disease caused by *H. vastatrix* in coffee plants and reduced symptoms in infected plants. Furthermore, PC chitosan at 0.05% concentration induced the activity of enzymes related to pathogenesis such as β-1,3 glucanase and peroxidase and increased the amount of total phenolic compounds. Of the four chitosans evaluated, PC chitosan resulted in lower intrinsic viscosity, lower viscosity average molecular weight, and (together with MC chitosan) a lower deacetylation degree. These properties of chitosan are important for its use in agriculture, specifically when used as an elicitor molecule. Therefore, PC chitosan could be used as an elicitor to increase the defense response of *Coffea arabica* against coffee rust.

## Figures and Tables

**Figure 1 polymers-13-01940-f001:**
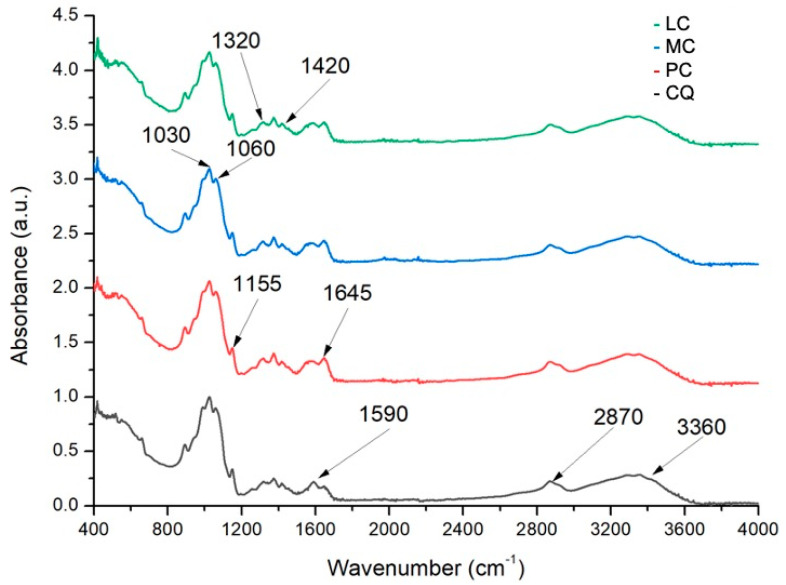
FTIR spectra of chitosan samples.

**Figure 2 polymers-13-01940-f002:**
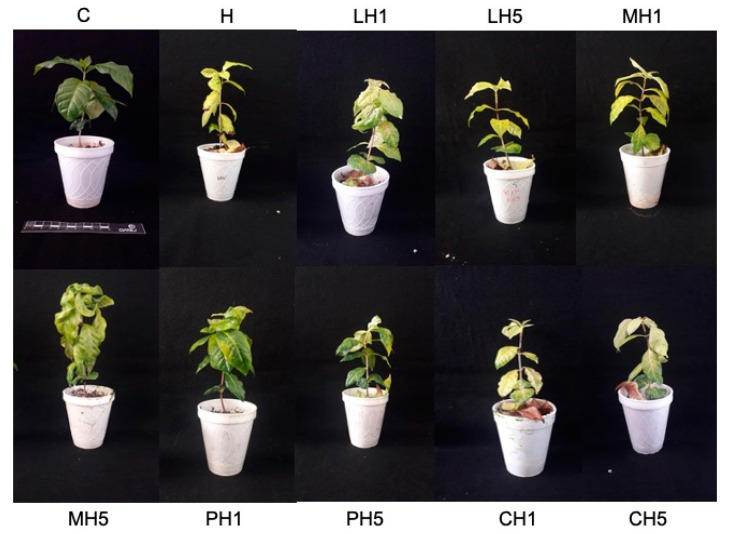
Day 80 of the biological effectiveness test. C: control plants: H: plants inoculated with *H. vastatrix;* LH1: plants treated with low-molecular-weight chitosan 0.01%, LH5: plants treated with low-molecular-weight chitosan 0.05%, MH1: plants treated with medium-molecular-weight chitosan 0.01%, MH5: plants treated with medium-molecular-weight chitosan 0.05%, PH1: plants treated with practical-grade chitosan 0.01%, PH5: plants treated with practical-grade chitosan 0.05%, CH1: plants treated with commercial food-grade chitosan 0.01%, CH5: plants treated with commercial food-grade chitosan 0.05%. All treatments were inoculated with *H. vastatrix*, except for C.

**Figure 3 polymers-13-01940-f003:**
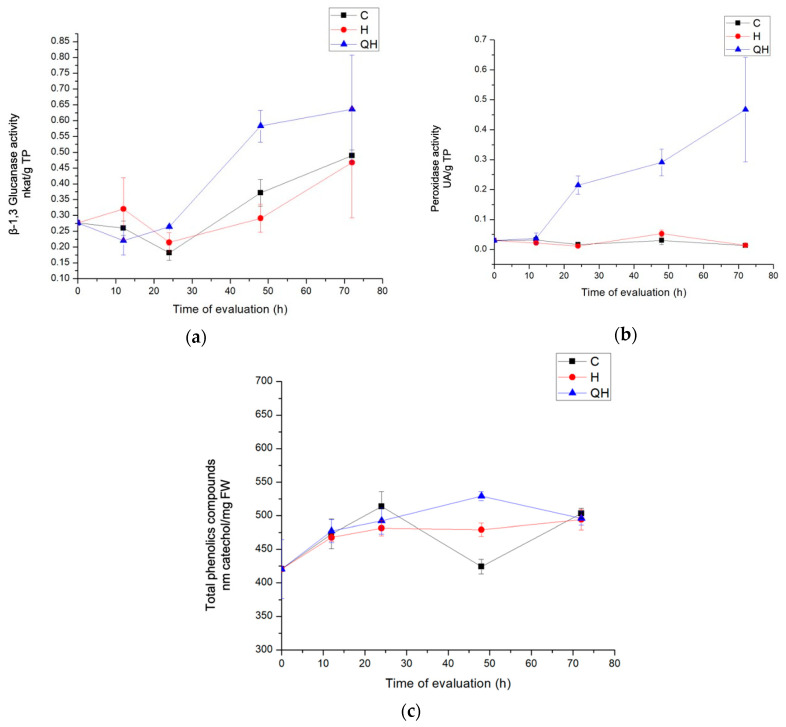
(**a**) Evaluation of β-1,3-glucanase activity in leaves of coffee plants, (**b**) Evaluation of peroxidase activity in leaves of coffee plants. (**c**) Evaluation of quantification of total phenolic compounds in leaves of coffee plants. The treatments evaluated were control (C) plants without any treatment and H plants inoculated with *H. vastatrix.* QH corresponds to plants treated with 0.05% practical-grade chitosan and inoculated with *H. vastatrix*; (0) corresponds to the time of inoculation with *H. vastatrix*; and 12, 24, 48, and 72 correspond to the hours after inoculation with *H. vastatrix*.

**Table 1 polymers-13-01940-t001:** Disease severity scale.

Value	Symptom
0	0% of leaf area damaged
1	Chlorotic spots on the leaf surface
2	2% of leaf area damaged
3	7% of leaf area damaged
4	20% of leaf area damaged
5	45% of leaf area damaged
6	More than 70% of the leaf area damaged

**Table 2 polymers-13-01940-t002:** Physicochemical properties of chitosans. Intrinsic viscosity, (η), viscosity average molecular weight (M*v*), and deacetylation degrees (DD) of chitosan.

Sample	[η]	M*v* (kD)	DD (%)
PC	593.5 ± 2.87	260.6 ± 23.96	77.22 ± 3.05
MC	829.9 ± 3.49	452.8 ± 36.99	76.81 ± 2.87
LC	704.9 ± 3.06	295.4 ± 54.86	79.69 ± 0.54
CQ	631.1 ± 10.90	314.4 ± 12.31	90.35 ± 2.57

Samples corresponds to practical-grade chitosan (PC), medium-molecular-weight chitosan (MC), low-molecular-weight chitosan (LC), and commercial high-density food-grade chitosan (CQ).

**Table 3 polymers-13-01940-t003:** Disease parameters.

Treatment *	Defoliation (%) *	Diseased Leaves (%) *	AUDPC *	Disease Incidence (%)
C	0 ± 0 ^a^	0 ± 0 ^a^	0 ± 0 ^a^	0
H	18.83 ± 5.81 ^b^	34.83 ± 8.23 ^bcd^	184 ± 2.58 ^e^	100
LH1	17.93 ± 8.24 ^b^	36.56 ± 7.01 ^cd^	166 ± 5.86 ^de^	100
LH5	24.67 ± 7.81 ^b^	39.05 ± 10.60 ^d^	180 ± 6.66 ^e^	100
MH1	20.51 ± 6.05 ^b^	33.12 ± 6.55 ^bcd^	179 ± 7.97 ^e^	100
MH5	16.72 ± 6.07 ^b^	38.69 ± 10.47 ^d^	178 ± 6.99 ^de^	100
PH1	15.58 ± 5.19 ^b^	37.50 ± 8.32 ^cd^	153 ± 2.41 ^cd^	100
PH5	19.55 ± 6.75 ^b^	22.23 ± 8.43 ^b^	76 ± 27.70 ^b^	80
CH1	15.88 ± 5.79 ^b^	42.64 ± 8.07 ^d^	79 ± 24.77 ^b^	90
CH5	14.47 ± 5.81 ^b^	24.47 ± 4.99 ^bc^	136 ± 20.97 ^c^	100

* Values presented are means with standard deviation. Values with the same letter are statistically equal, and values with different letters show statistically significant differences at the 95% Tukey confidence level (*p* < 0.05).

## Data Availability

The data presented in this study are available on request from the corresponding author.

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
