# Peer review of "Evaluation of the Physicochemical Properties of Chitosans in Inducing the Defense Response of Coffea arabica against the Fungus Hemileia vastatrix"

_polymers, 2021, doi:10.3390/polym13121940_

Round 1
Reviewer 1 Report
Julio César López-Velázquez, José Nabor Haro-González, Soledad García-Morales, Hugo Espinosa-Andrews, Diego Eloyr Navarro-López, Mayra Montero-Cortés5 and Joaquín Alejandro Qui-Zapata have presented evaluation of chitosan with different physicochemical properties for the control of H. vastatrix in coffee plants.
Based on my pondered analysis, the positive point of this manuscript is the fact that the introduction is well described, the experiment was presented comprehensively and adequate methods were used to analyze the samples and achieve the assumed goal, the results are supported by the literature, with appropriate citations provided.
However, there are some concerns:
- The beginning of the abstract should be reconstructed – lines 21-24. Instead of this, please highlight the purpose of your study.
- All significant differences should be written in superscript – Table 3.
- Please add the standard deviation to the results in Table 2.
- I have some doubts regarding the English style and language.
The weakest part of the article is conclusion.
Reviewer 2 Report
The aim of the study was to evaluate the effect of chitosan (with different physicochemical parameters) on the induction of Coffea arabica defense against Hemileia vastatrix. The subject of the manuscript is interesting, although there are many publications previously published that demonstrate the properties of chitosan as a molecule that has the ability to induce a response against phytopathogens. The article would also be more interesting if the changes in the activity of all defense enzymes and compounds included in the cell wall were also examined. The article is written in an interesting way, although in my opinion the discussion needs to be corrected. The analytical methods used in the work are standard, the references cited in the work are also appropriate.
Detailed comments:
L157- Capital letter at the beginning of the sentence ..”Those plants…
L228 - In determining the content of polyphenols, it should be added that the reactions were carried out in the presence of sodium carbonate
Why is it stated in the methodology of total polyphenol determination that their content was defined as nmol catechol / g FW. And in graph number 3C the results are in A650 mg / FW? If this is a mistake, it needs to be corrected.
In part of the discussion, many research results that were presented in previously published articles are given. However, this makes this part of the manuscript more of an introduction than a discussion of the results obtained. In this chapter, I recommend that you add a detailed discussion of the results obtained, especially the influence of the type of chitosan on the enzymatic activity and content of polyphenols.
